

Tropical Continental Downdraft Characteristics: Mesoscale Systems versus Unorganized

Convection

Kathleen A. Schiro[1] and J. David Neelin[1]

[1]*Department of Atmospheric and Oceanic Sciences, University of California Los Angeles, Los*

*Angeles, CA, USA*

Corresponding Author: Kathleen A. Schiro, Department of Atmospheric and Oceanic Sciences,

University of California, Los Angeles, Box 951565, Los Angeles, CA 90095.  E-mail:

kschiro@atmos.ucla.edu

**Abstract**

Downdrafts and cold pool characteristics for mesoscale convective systems (MCSs) and

isolated, unorganized deep precipitating convection are analyzed using multi-instrument data
from the GOAmazon campaign. For both MCSs and isolated cells, there are increases in column
water vapor (CWV) observed in the two hours leading the convection and an increase in wind
speed, decrease in surface moisture and temperature, and increase in relative humidity coincident
with system passage. Composites of vertical velocity data and radar reflectivity from a radar
wind profiler show that the downdrafts associated with the sharpest decreases in surface
equivalent potential temperature ($\theta_e$) have a probability that increases towards lower levels
below the freezing level. Both MCSs and unorganized convection show similar mean downdraft
magnitudes and probabilities with height.  This is consistent with thermodynamic arguments: if
$\theta_e$ were approximately conserved following descent, it would imply that a large fraction of the
air reaching the surface originates at altitudes in the lowest 2 km, with probability of lower $\theta_e$
dropping exponentially. Mixing computations suggest that, on average, air originating at heights
greater than 3 km must undergo substantial mixing, particularly in the case of isolated cells, to
match the observed cold pool $\theta_e$, likewise implying a low typical origin level.  Precipitation
conditionally averaged on decreases in surface equivalent potential temperature ($\Delta\theta_e$) exhibits a
strong relationship because the largest $\Delta\theta_e$ values are associated with high probability of
precipitation.  The more physically motivated conditional average of $\Delta\theta_e$ on precipitation levels





off with increasing precipitation rate, bounded by the maximum difference between surface $\theta_e$
and its minimum in the profile aloft. Precipitation values greater than about 10 mm h$^{-1}$ are
associated with high probability of $\Delta\theta_e$ decreases. Robustness of these statistics observed across
scales and regions suggests their potential use as model diagnostic tools for the improvement of
downdraft parameterizations in climate models.
**1 Introduction**
Convective downdrafts involve complex interactions between dynamics,
thermodynamics, and microphysics across scales. They form cold pools, which are evaporatively
cooled areas of downdraft air that spread horizontally and can initiate convection at their leading
edge (Byers and Braham 1949; Purdom 1976; Wilson and Schreiber 1986; Rotunno et al. 1988;
Fovell and Tan 1998; Tompkins 2001; Khairoutdinov and Randall 2006; Lima and Wilson 2008;
Khairoutdinov et al. 2009; Boing et al. 2012; Rowe and Houze 2015). The boundary between the
cold pool and the surrounding environmental air, known as the outflow boundary or gust front, is
the primary mechanism for sustaining multi-cellular deep convection (e.g. Weisman and Klemp
1986). It has also been shown to trigger new convective cells in marine stratocumulus clouds
(Wang and Feingold 2009; Terai and Wood 2013) and in trade-wind cumulus (Zuidema et al.
2011; Li et al. 2014). Downdrafts also have implications for new particle formation in the
outflow regions, which contribute to maintaining boundary layer CCN concentrations in
unpolluted environments (Wang et al. 2016).
Precipitation-driven downdrafts are primarily a result of condensate loading and the
evaporation of hydrometeors in unsaturated air below cloud base (e.g. Houze 1993), with
evaporation thought to be the main driver (Knupp and Cotton 1985; Srivastava 1987). It was
originally suggested by Zipser (1977) that the downdrafts in the convective part of a system,
referred to in the literature as convective-scale downdrafts, are saturated and the downdrafts in
the trailing stratiform region (referred to as mesoscale downdrafts) are unsaturated. Studies with
large-eddy simulations (LES; Hohenegger and Bretherton 2011; Torri and Kuang 2016) indicate,
however, that most convective downdrafts are unsaturated, consistent with evidence that the
evaporation of raindrops within the downdraft likely does not occur at a sufficient rate to
maintain saturation (Kamburova and Ludlam 1966).





More recently, studies have shown the importance of downdraft parameters in
maintaining an accurate simulation of tropical climate in global climate models (GCMs;
Maloney and Hartmann 2001; Sahany and Nanjundiah 2008; Del Genio et al. 2012;
Langenbrunner and Neelin 2017). Accurate simulation of MCSs in continental regions (Pritchard
et al. 2011) was also shown to be sensitive to downdraft–boundary layer interactions, with
significantly improved representation of MCS propagation in the central US once such
interactions were resolved. Additionally, representing the effects of downdrafts and cold pools in
models has been shown to have positive effects on the representation of the diurnal cycle of
precipitation (Rio et al. 2009; Schlemmer and Hohenegger 2014).
This study aims to characterize downdrafts in a comprehensive way in the Amazon for
both isolated and mesoscale convective systems, and to provide useful guidance for downdraft
parameterization in GCMs. Data from the DOE–Brazil Green Ocean Amazon (GOAmazon)
campaign (2014–2015; Martin et al. 2016) provides an unprecedented opportunity to evaluate
downdraft characteristics in the Amazon with sufficiently large datasets for quantifying robust
statistical relationships describing leading order processes for the first time. Relationships
explored previously, primarily in tropical oceanic or mid-latitude regions, such as time
composites of wind and thermodynamic quantities relative to downdraft precipitation, are also
revisited and compared to our findings over the Amazon. Downdrafts in MCSs and isolated cells
are compared to inform decisions concerning their unified or separate treatment in next
generation models. The effect of downdrafts on surface thermodynamics and boundary layer
recovery are examined, and the origin height of the downdrafts explored, combining inferences
from radar wind profiler data for vertical velocity and thermodynamic arguments from simple
plume models. Lastly, statistics describing cold pool characteristics at the surface are presented
and discussed for possible use as model diagnostics.
**2 Data and Methods**
Surface meteorological values (humidity, temperature, wind speed, precipitation) were
obtained from the Aerosol Observing meteorological station (AOSMET) at the DOE ARM site
in Manacapuru, Brazil, established as part of the GOAmazon campaign (site T3; ARM Climate
Research Facility 2013a). The record used in this study spans 10 Jan 2014–20 Oct 2015. Values
in this study are averaged at 30-min intervals. Equivalent potential temperature is computed
following Bolton (1980). Sensible and latent heat fluxes (30-min) are derived from eddy



correlation flux measurements obtained with the eddy covariance technique involving correlation
of the vertical wind component with the horizontal winds, temperature, water vapor density, and
carbon dioxide concentration (ARM Climate Research Facility 2014). A fast-response, three-
dimensional sonic anemometer provides the wind components and speed of sound, while water
vapor density is from an open-path infrared gas analyzer. Surface flux data from 03 Apr 2014–20
Oct 2015 are used here, with periods of missing and unreliable data excluded, as flagged by
ARM.

Thermodynamic profiles are obtained from radiosonde measurements within 6 h of a

convective event (ARM Climate Research Facility 2013b). Radiosondes are launched at
approximately 01:30, 7:30, 13:30, and 19:30 LT each day, with occasional radiosondes at 10:30
LT in the wet season. Profiles of vertical velocity and radar reflectivity are obtained from a 1290
MHz radar wind profiler (RWP) reconfigured for precipitation modes. It has a beam width of $6^o$
(~ 1 km at 10 km AGL), a vertical resolution of 200 m, and a temporal resolution of 5 seconds
(Giangrande et al. 2016).

Precipitation data at 25 km and 100 km, as well as convection classifications, are derived

from an S-Band radar located approximately 67 km to the northeast of T3 at the Manaus Airport.
Composite constant altitude low-level gridded reflectivity maps (constant altitude plan position
indicators, CAPPIs) were generated, and the radar data were gridded to a Cartesian coordinate
grid with horizontal and vertical resolution of 2 km and 0.5 km, respectively (ARM Climate
Research Campaign Data, C. Schumacher, 2015). Rain rates were obtained from the 2.5 km
reflectivity using the reflectivity-rain rate (Z-R) relation $Z=174.8R^{1.56}$ derived from disdrometer
data (ARM Climate Research Campaign Data, C. Schumacher, 2015). The spatially averaged
rainfall rate over a 25 km and 100 km grid box were used in this study. The center of the 100 km
grid box is shifted slightly to the right of center with respect to the T3 site due to reduced data
quality beyond a 110 km radius.

All convective events used in this study meet the following criteria: producing

downdrafts that create a subsequent drop in $\theta_e$ at the surface of less than $-5^o$ C in a 30-min period
and having precipitation rates exceeding 10 mm h$^{-1}$ in that same period. These criteria were
chosen to examine the most intense downdraft events with the most well-defined vertical
velocity signatures in the RWP data. Only data for events with complete vertical velocity data



coverage over the 1 h period spanning the passage of the convective cells and centered around
the maximum precipitation were composited and evaluated.

Isolated convective cells were identified by S-Band composite reflectivity, as in Fig. 1,

and are defined as being less than 50 km in any horizontal dimension (contiguous pixels with
reflectivity > 30 dBZ) with a maximum composite reflectivity of greater than or equal to 45 dBZ.
Following the criteria defined above, this resulted in the selection of 11 events, all of which were
in the late morning or afternoon hours between 11:00 and 18:00 LT. Mesoscale convective
systems follow the traditional definition of regions of contiguous precipitation at scales of 100
km or greater (contiguous pixels with reflectivity > 30 dBZ) in any horizontal dimension (e.g.
Houze 1993; Houze 2004). All of the events sampled are characterized by a leading edge of
convective cells with a trailing stratiform region (Fig. 1), which is the most common MCS type
(Houze et al. 1990). The above criteria yielded 17 events: 11 in the late morning and early
afternoon hours (11:00-18:00 LT) and 6 in the late evening/early morning hours (22:00-11:00
LT).

In Sect. 6, statistics are presented using nearly the entire two-year timeseries of

meteorological variables at the GOAmazon site, as well as 15 years of data (1996–2010) from
the DOE ARM site at Manus Island in the tropical western Pacific. One-hour averages are
computed in $\Delta\theta_e$ and precipitation.
**3 Surface Thermodynamics**

Composites of surface meteorological variables are displayed in Fig. 2 for the 11 isolated

cellular deep convective events coinciding with drops in equivalent potential temperature of -5$^\circ$C
or less and precipitation rates greater than 10 mm h$^{-1}$ (see Sect. 2). The composites are centered 3
h before and after the time marking the beginning of the sharpest decrease in surface $\theta_e$. All
differences quoted are the differences in values between the maximum and minimum values
within the 1 h timeframe of convective cell passage, unless noted otherwise. All timeseries
averaged in the composites are shifted to the mean value at 0.5 h, the timestep immediately
following the minimum $\Delta\theta_e$, and error bars on the composites are +/- 1 standard deviation with
respect to 0.5 h.

In the two hours leading the convection, the CWV increases by 4.3 mm. Values of $\theta_e$ are

353.6 K on average before passage of the cell. An hour after the passage, the $\theta_e$ value drops by



an average 8.9° to an average value of 344.7 K. Since the isolated convective cells observed
occur in the daytime hours, the relative humidity is seen to drop steadily throughout the 3 h
period leading the convection following the rise in temperatures with the diurnal cycle. Once the
cell passes, RH values rise to 81.6%, which indicates that the downdrafts are sub-saturated when
they reach the surface. Temperatures drop by 4.4° C to 24.9° C, which is less of a drop in
temperature than observed over mid-latitude sites (see Table 2 in Engerer et al. 2008 for a review
of mid-latitude case studies) and specific humidity drops by 1.1 g kg$^{-1}$ to 16.0 g kg$^{-1}$. Wind
speeds reach 5.5 m s$^{-1}$ on average, consistent with previous studies that document strong
horizontal winds associated with the leading edges of cold pools (e.g. Fujita 1963; Wakimoto
1982), but are lower than the observed values for mid-latitude storms (Engerer et al. 2008).
Additionally, surface pressure often increases with the existence of a cold pool and is referred to
as the meso-high (Wakimoto 1982). Here, it increases marginally by 0.8 hPa, but this value is
much less than the typical values observed in mid-latitudes (e.g. Goff 1976; Engerer et al. 2008).
Lastly, 63% of the temperature and moisture depleted by the downdraft recovers within two
hours of cell passage, with moisture recovering more quickly and by a greater percentage than
temperature.
Complementary to those in Fig. 2, composites of surface meteorological variables are
shown in Fig. 3 for the 17 MCSs with surface $\theta_e$ depressions of -5° C or less and coincident
precipitation rates of 10 mm h$^{-1}$ or greater. On average, the environment is more humid, as is
seen in the CWV composite. Values of $\theta_e$ leading the passage of MCSs are a few degrees lower
than the $\theta_e$ values leading the isolated cells. This is mostly due to lower surface temperatures.
The precipitation occurs over a longer period than in the cases of isolated cells, as there is
stratiform rain trailing the leading convective cells. The stratiform rain and associated
downdrafts also sustain the cooling and drying of the near surface layers for many hours lagging
the precipitation maximum. Column water vapor values leading the MCSs are slightly higher on
average than observed leading the isolated cells, with an average maximum value of 59.8 mm.
The relative humidity maximum in the cold pool is 90.2% ($\Delta RH = 14.2\%$), the specific humidity
minimum is 15.5 g kg$^{-1}$ ($\Delta q = 1.7$ g kg$^{-1}$), and the temperature minimum is 22.9° C ($\Delta T = 4.7°$ C),
with winds gusting to an average of 6.3 m s$^{-1}$ with the passage of the leading convective cells.
The cold pools are thus cooler, drier, and nearer to saturation for the MCSs than for the isolated



cells. It is worth noting that these statistics for MCSs are not greatly affected by the inclusion of
nighttime events; composites for afternoon only MCSs yield similar results.

Overall, the environments in which MCSs live are moister, they have colder, drier cold

pools that are nearer to saturation, the winds at their leading edges are gustier, and their boundary
layers recover more slowly than for isolated cells.
**4 Downdraft Origin and the Effects of Mixing**

Many previous studies of moist convective processes use $\theta_e$ as a tracer since it is

conserved in the condensation and evaporation of water and for dry and moist adiabatic
processes (e.g., Emanuel 1994). Tracing surface $\theta_e$ to its equivalent value aloft has been used in
many studies of tropical convection to examine potential downdraft origin heights (e.g. Zipser
1969; Betts 1973, 1976; Betts and Silva Dias 1979; Betts et al. 2002). This assumes that
downdraft air conserves $\theta_e$ to a good approximation and that downdraft air originates at one
height above ground level. Neither of these assumptions is likely to be true, as mixing is likely
occurring between the descending air and the environmental air and thus originating from
various levels. However, it can provide a useful reference point for further considerations.

We examine the mean $\theta_e$ profiles for MCSs and isolated cells, conditioned on the

existence of a substantial drop in $\theta_e$ and precipitation rates above a threshold value, to place
bounds on mixing and downdraft origin with simple plume computations. Matching the
minimum $\theta_e$ value observed at the surface following the passage of convection to the minimum
altitude at which those values are observed yields 1.3 km for isolated cells (left panel, Fig. 4) and
2.0 km for MCSs (right panel, Fig. 4). Again, this assumes that $\theta_e$ is conserved and that the air
originates at one altitude. If instead we assume that substantial mixing occurs with the
surrounding environment and that air originates at multiple levels in the lower troposphere, it
would be plausible for more of the air reaching the surface to originate at altitudes greater than
1.3 and 2 km for isolated cells and MCSs, respectively. This has been alluded to in previous
studies (e.g. Zipser, 1969; Gerken et al. 2016), which provide evidence that air originates in the
middle troposphere.

To examine this, we mix air from above the altitude where the $\theta_e$ matched the surface

value (shown in the composites in Figs. 2 and 3) downward towards the surface, varying the
entrainment rate (constant with pressure). To start, we use a mixing of 0.001 hPa$^{-1}$, as this is the



constant entrainment value used in Holloway and Neelin (2009) and Sahany et al. (2012), which
produced realistic updraft buoyancy profiles over tropical oceans. For the MCS case, it is
plausible that a downdraft could originate at a height of 2.3 km given this rate of mixing to reach
the surface with characteristics given by Fig. 3. (Note that there is a spread in surface values and
profile characteristics, but for simplicity we use mean values.) If instead the air were coming
from the level of minimum $\theta_e$, an assumption similar to that made by many downdraft
parameterizations (e.g. Zhang and McFarlane 1995; Tiedke 1989; Kain and Fritsch 1990),
mixing would need be 2.5 times greater. For the isolated cells, mixing rates appear to need to be
much greater in order to produce results consistent with those seen at the surface. If we start out
at 0.0025 hPa$^{-1}$, the rate sufficient for a minimum $\theta_e$ origin for the MCSs, this only yields an
origin height of 1.5 km. If instead we assume the air originates near the level of minimum $\theta_e$,
mixing would need to be at least 0.006 hPa$^{-1}$. For reference, in the Tiedke and Zhang-McFarlane
schemes, downdrafts mix with environmental air at a rate nearly double the rate of mixing in
updrafts, which in the Tiedke scheme is 2 x 10$^{-4}$ m$^{-1}$. This is similar to 0.0025 hPa$^{-1}$ in pressure
coordinates in the lower troposphere.

To summarize, this analysis is suggestive of bounds on mixing coefficients for downdraft

parameterizations. Downdrafts would need to mix less substantially through the lower
troposphere for MCSs than isolated cells to draw down air that matched the observed
characteristics at the surface, and the rate of mixing needed to bring air down from the level of
minimum $\theta_e$ would be 2.5 times greater for isolated cells than for the MCSs. In Sections 5 and 6,
we provide a complementary probabilistic perspective on levels of origin.
**5 Vertical Velocity and Downdraft Probability**

Figure 5 composites reflectivity (Z), vertical velocity (w), and the probability of

observing downdrafts (w < 0 m s$^{-1}$) for the 11 cases of isolated cellular convection meeting the
minimum $\Delta\theta_e$ criteria of -5$^{\circ}$ C and minimum precipitation criteria of 10 mm h$^{-1}$. Time 0 is the
time right before the sharpest decrease in $\theta_e$, repeated from Fig. 2 in the top panel, and maximum
precipitation. A 3 h window is composited for reference, but the interval of primary interest is
the 1 h window within which the minimum $\Delta\theta_e$ and maximum precipitation are observed. To
highlight the interval of interest, the 1 h intervals leading and lagging this period are masked out.



The drop in $\theta_e$ is coincident with the passage of the isolated cell and its main updraft and
precipitation-driven downdraft. Mean reflectivity exceeding 40 dBZ is observed during this
period, as are strong updrafts in the middle-upper troposphere. The cell then dissipates and/or
moves past the site within an hour. A downdraft is observed directly below and slightly trailing
the updraft core. This is the downdraft that is associated with the largest drop in surface $\theta_e$. As is
suggested in the literature, these are mainly driven by condensate loading and evaporation of
precipitation and are negatively buoyant. The probability of observing negative vertical velocity
(threshold < 0 m s$^{-1}$) within the 30 minutes of observed maxima in the absolute value of $\Delta\theta_e$ and
precipitation is highest in the lower troposphere (0-2 km), consistent with precipitation-driven
downdrafts observed in other studies (Sun et al. 1993; Cifelli and Rutledge 1994).
There is also a high probability of downdrafts in air near the freezing level (masked out in
the vertical velocity retrievals, as there is large error associated with retrievals near the freezing
level; Giangrande et al. 2016). It appears likely, however, that these downdrafts are
discontinuous in height more often than not, as high probabilities are not observed coincidentally
in the lowest levels beneath these downdrafts. These mid-upper level downdrafts are documented
in previous studies of MCSs, suggesting that they form in response to the pressure field (e.g.
Biggerstaff and Houze 1991), can occur quite close to the updraft (Lily 1960; Fritsch 1975), and
are positively buoyant (Fovell and Ogura 1988; Jorgensen and LeMone 1989; Sun et al. 1993).
These motions produce gravity waves in upper levels, as is discussed in Fovell et al. (1992).
Figure 6 shows the same composites for the 17 MCSs observed. They, too, have high
reflectivity (mean > 40 dBZ) in the 30 minutes coincident with the minimum $\theta_e$ and a defined
updraft extending up to the upper troposphere. Downdrafts occurring coincident with the
minimum $\theta_e$ are observed directly below the updraft signature in the mean vertical velocity
panel, and the probabilities are greatest below the freezing level. There is also evidence of
mesoscale downdrafts in the trailing stratiform region of the MCSs, which Miller and Betts
(1977) suggest are more dynamically driven than the precipitation-driven downdrafts associated
with the leading-edge convection. These sustain the low $\theta_e$ air in the boundary layer for hours
after the initial drop, observed in Fig. 3. Vertical motions in the stratiform region are weaker than
in the convective region, and on average, as in Cifelli and Rutledge (1994), rarely exceed 1 m s$^{-1}$.
Figure 7 is a concise summary of the results presented in Figs. 5 and 6, showing the mean
vertical velocity within the 30-min of sharpest $\Delta\theta_e$ for MCSs and isolated cells. Previous studies



using radar wind profilers have shown mean updraft and downdraft strength increases with
height (May and Rajopadhyaya 1999; Kumar et al. 2015; Giangrande et al. 2016), consistent
with our results here for both isolated and organized deep convection. The corresponding mean
probability is shown in the right panel. The probability of downdrafts for both isolated cells and
MCSs increases nearly linearly towards the surface below the freezing level. Thus, the behavior
in the lowest 3 km summarizes our results from the previous two figures and suggests that
downdrafts accumulate air along their descent, analogous to mixing. Probabilities, which can be
interpreted loosely as convective area fractions (Kumar et al. 2015; Giangrande et al. 2016), are
also largest below the freezing level for downdrafts and in the 3-7 km region for updrafts. The
probability and vertical velocity for both MCSs and isolated cells correspond to mass flux
profiles that increase nearly linearly throughout the lower troposphere for updrafts and that
decrease nearly linearly throughout the lower troposphere for downdrafts, as seen in Giangrande
et al. (2016) over a broader range of convective conditions.
These results suggest that in most downdrafts, a substantial fraction of the air reaching
the surface originates in the lowest 3 km within both organized and unorganized convective
systems. Several observational studies corroborate the evidence presented here that a majority of
the air reaching the surface in deep convective downdrafts originates at low-levels (Betts 1976;
Barnes and Garstang 1982; Betts et al. 2002). Betts 1976 concluded that the downdraft air
descends approximately only the depth of the subcloud layer (~150 mb). Betts et al. (2002) cited
a range of 765-864 hPa for the first levels at which the surface $\theta_e$ values matched those of the air
aloft. Additionally, there are many modeling studies that provide evidence of these low-level
origins (Moncrieff and Miller, 1976; Torri and Kuang, 2016). Recently, Torri and Kuang (2016)
used a Lagrangian particle dispersion model to show that precipitation-driven downdrafts
originate at very low levels, citing an altitude of 1.5 km from the surface. These conclusions are
consistent with our results here, suggesting that downdraft parameterizations substantially weight
the contribution of air from the lower troposphere (e.g. with substantial mixing, modifying height
of downdraft origin).
**6 Relating Cold Pool Thermodynamics to Precipitation**
As seen in previous sections, the passage of both organized and unorganized convective
cells can lead to substantial decreases in $\theta_e$ resulting mainly from precipitation-driven





downdrafts formed from the leading convective cells. In this section, we search for robust
statistical relationships between key thermodynamic variables for potential use in improving
downdraft parameterizations in GCMs. These statistics differ from those presented in Figs. 2-7,
as these statistics are not conditioned on convection type and sample both precipitating and non-
precipitating points within the timeseries analyzed. All data available at the surface
meteorological station during the GOAmazon campaign from 10 Jan 2014–20 Oct 2015 are
included in these statistics.

The first of these statistics conditionally averages precipitation rate by $\Delta\theta_e$ (Fig. 8),

variants of which have been discussed in previous studies (Barnes and Garstang 1982; Wang et
al. 2016). Our statistics mimic those shown in previous work relating column-integrated moisture
to deep convection over tropical land (Schiro et al. 2016) and ocean (Neelin et al. 2009;
Holloway and Neelin 2009). The direction of causality in the CWV-precipitation statistics,
however, is the opposite of what is presented here. CWV is thought to primarily be the cause of
intense precipitation and deep convection, while here the $\Delta\theta_e$ observed is a direct result of the
precipitation processes and associated downdraft. Nevertheless, examining the distribution of
$\Delta\theta_e$ observed at the surface and magnitudes of the rain rates associated with the highest drops in
$\Delta\theta_e$ across different regions in the tropics can place bounds on downdraft behavior. We will also
conditionally average $\Delta\theta_e$ by precipitation rate, a more physically consistent direction of
causality.

Figure 8 shows precipitation rate binned by $\Delta\theta_e$ for in-situ precipitation and radar

precipitation. Bins are $1^o$ C in width and precipitating events are defined as having rain rates
greater than 1 mm h$^{-1}$. These statistics mainly suggest that any substantial decrease in $\theta_e$ at the
surface occurs coincidently with heavy precipitation, which is particularly evident from the sharp
increase in probability of precipitation (middle panel). The width of the distribution of
precipitating points is of greatest interest here. The distribution of precipitating points peaks just
shy of a $\Delta\theta_e$ of $0^o$ C, indicating that most precipitation events have low rain rates and do not
occur coincidently with an appreciable drop in $\Delta\theta_e$. The frequency of precipitation drops off
roughly exponentially towards lower $\Delta\theta_e$. An interesting feature is the lower bound observed in
the $\Delta\theta_e$ near -15$^o$ C. Examining mean profiles in Fig. 5 show that, on average, this value of -15$^o$
C would be consistent with air originating from the level of minimum $\theta_e$ and descending
undiluted to the surface. The frequency of observing these values suggests that air very rarely





reaches the surface from these altitudes (3 km or higher) undiluted.  The $\theta_e$ probability
distribution is consistent with the results of Sect. 5, indicating that the probability of air from a
given level of origin reaching the surface increases toward the surface through the lowest 3 km.
S-Band radar data are averaged in 25 km and 100 km grid boxes surrounding the
GOAmazon site to examine the precipitation-$\Delta\theta_e$ relation with model diagnostics in mind (Fig.
8). Out to 25 km, the statistics are very similar to those observed using in situ precipitation.
Theoretical (Romps and Jevanjee 2015), modeling (Tompkins 2001; Feng et al. 2015), and
observational (Feng et al. 2015) studies have all examined typical sizes of cold pools, which are
on the order of 25 km in diameter for any one cell. Cold pools can combine, however, to form a
larger, coherent mesoscale-sized cold pool (radius of 50 km or greater), as is commonly
associated with mesoscale convective systems (Fujita 1959; Johnson and Hamilton 1988).
Therefore, it is likely that our use of the in situ $\Delta\theta_e$, assuming cold pool properties are somewhat
homogeneous in space, is appropriate for scales up to 25 km. Beyond this scale, it is likely that
the $\Delta\theta_e$ would be smoothed by averaging, particularly for the smaller isolated cells, as would
precipitation. For 100 km, the precipitation is smoothed by averaging, which would likely
degrade further if information of 100 km mean surface thermodynamics were available. This
suggests that comparing these statistics to those produced with model output for diagnostic
purposes would yield a narrower range of $\Delta\theta_e$ and lower conditionally averaged rain rates.
Figure 9 shows remarkable similarity in these statistics when comparing across regions to
a DOE ARM site at Manus Island in the tropical western Pacific. As $\Delta\theta_e$ decreases, in situ
precipitation rates sharply increase. The distributions, as well as the steepness and locations of
the pickups, are remarkably consistent. Again, the sharpness of these curves is a result of the
strongest precipitation events coinciding with the strongest decreases in $\theta_e$, shown in the middle
panels in Fig. 9, where the probability of observing precipitation is greatest at lower $\Delta\theta_e$.
It is then of interest to see if for a given precipitation rate we can expect a particular $\Delta\theta_e$,
as this is the proper direction of causality. Figure 10 conditionally averages $\Delta\theta_e$ by precipitation
rate (1-h averages). The maximum $\Delta\theta_e$ within a 3-h window of a given precipitation rate is
averaged to minimize the effects of local precipitation maxima occurring slightly before or after
the minimum in $\Delta\theta_e$. Comparing Fig. 9 and Fig. 10 shows that there can be strong precipitation
events without large, corresponding decreases in surface $\theta_e$, but that large decreases in surface $\theta_e$
are almost always associated with heavy precipitation.





Beyond about 10 mm h$^{-1}$ there is high probability of observing large, negative $\Delta\theta_e$ and an
apparent limit in mean $\theta_e$ decreases with rain rate. This makes physical sense, as discussed
above (see also Barnes and Garstang 1982), since cooling is limited by the maximum difference
between the surface $\theta_e$ and the $\theta_e$ minimum aloft. The average $\Delta\theta_e$ for rain rates exceeding 10
mm h$^{-1}$ is about -5$^{\rm o}$C for the Amazon and -4$^{\rm o}$C for Manus Island. This statistic could be of use in
constraining downdraft parameters to be consistent with surface cooling and drying observed in
nature. There are still, however, open questions about scale dependence and how much cooling
or drying should be observed for varying space and time scales. This result is likely applicable to
GCM grid scales of 0.25$^{\rm o}$ or less, as is suggested from the results in Fig. 9, but would be of lesser
magnitude at scales more comparable to typical GCM grids (100 km or greater). Overall, if
convective precipitation is present in a GCM grid, a corresponding $\Delta\theta_e$ should result within a
range consistent to those observed here, subject to scale dependence.
**7 Conclusions**
Convective events sampled during the GOAmazon campaign compare downdraft
characteristics between MCSs and isolated cells and examine their respective effects on surface
thermodynamics. All events included in the analysis passed directly over the GOAmazon site
with minimum precipitation rates of 10 mm h$^{-1}$ and $\Delta\theta_e$ less than or equal to -5$^{\rm o}$C. The isolated
events sampled occurred in the afternoon hours only and were characterized by average
decreases of 1.1 g kg$^{-1}$ in specific humidity, 3.9$^{\rm o}$C in temperature, and 8.0$^{\rm o}$C in $\theta_e$, with an
increase of 5.5 m s$^{-1}$ in wind speed at the surface. More than half of the deficit in $\theta_e$ observed
with the passage of the cells recovers within 2 h, on average, with the moisture recovering faster
than temperature and a larger fraction of the total $\theta_e$ recovered. MCSs show similar decreases in
temperature (3.7$^{\rm o}$C) but larger decreases in moisture (1.5 g kg$^{-1}$) and thus $\theta_e$ (9.1$^{\rm o}$C) at the
surface. The $\theta_e$ recovers more slowly for MCSs due to the mesoscale downdrafts and associated
precipitation in their trailing stratiform regions.
Vertical velocity profiles from a radar wind profiler show that the probability of
observing downdraft air during the 30 minutes of observed minimum $\Delta\theta_e$ increases with
decreasing height in the lowest 3 km for both isolated cells and MCSs. This vertical structure of
the downdraft probability is consistent with negative vertical velocities originating at various
levels within this layer and continuing to the surface. Considering complementary



thermodynamic arguments, without mixing, profiles of $\theta_e$ suggest that origin levels at average
altitudes of 1.3 and 2 km for isolated cells and MCSs, respectively, would be consistent with
average cold pool $\theta_e$ for these cases. A minimum in $\theta_e$ is observed between 3 and 7 km, on
average, so for air to originate above 3 km, simple plume calculations suggest that downdrafts in
MCSs would have to be mixing with environmental air at an approximate rate of 0.0025 hPa$^{-1}$
along descent and at a rate roughly 2.5 times greater (0.006 hPa$^{-1}$) for isolated cells. This would
imply mass entering the downdraft throughout the lowest few kilometers. Overall the vertical
velocity and thermodynamic constraints are consistent in suggesting a spectrum of downdraft
mass origin levels throughout the lowest few kilometers.

Robust statistical relationships between $\Delta\theta_e$ and precipitation are examined from nearly

two years of data at the GOAmazon site and 15 years of data at the DOE ARM site at Manus
Island in the tropical western Pacific. We conditionally average precipitation by $\Delta\theta_e$, similar to
the statistics of precipitation conditioned on a thermodynamic quantity we consider for
convective onset statistics. Here, however, the most likely direction of causality differs in that
the $\theta_e$ drop is caused by the downdraft that delivers the precipitation (as opposed to the
thermodynamic profile providing convective available potential energy for an updraft). For in
situ precipitation, the conditional average precipitation exhibits a sharp increase with decreasing
$\Delta\theta_e$, which is similar in magnitude over land and ocean, reaching roughly 10 mm hour$^{-1}$ at a $\Delta\theta_e$
of -10° C. For area-averaged precipitation on scales typical of GCM grids, precipitation
magnitude is lower for strong, negative $\Delta\theta_e$, consistent with the points with large $\Delta\theta_e$ occurring
at localized downdraft locations within a larger system with smaller area-average precipitation.
The probability distributions of $\Delta\theta_e$ (for precipitating and non-precipitating points) over land and
ocean are also remarkably similar. Distributions show exponentially decreasing probability with
decreasing $\Delta\theta_e$, providing additional evidence that downdraft plumes originating in the lowest
levels are orders of magnitude more likely than plumes descending with little mixing from the
height of minimum $\theta_e$. Conditionally averaging $\Delta\theta_e$ by precipitation (the most likely direction of
causality) suggests an average limit in $\Delta\theta_e$ of -4° C to -5° C given high precipitation typical of
downdraft conditions. The corresponding 90[th] percentile yields $\Delta\theta_e$ of roughly -10° C, consistent
with results obtained from composting strong downdrafts. The robustness of these statistics over
land and ocean, and to averaging in space at scales appropriate to a typical GCM resolution,





suggests possible use of these statistics as model diagnostic tools and observational constraints
for downdraft parameterizations.
**Acknowledgments**

The U.S. Department of Energy Atmospheric Radiation Measurement (ARM) Climate

Research Facility GOAmazon and Tropical West Pacific field campaign data were essential to
this work. This research was supported in part by the Office of Biological and Environmental
Research of the U.S. Department of Energy Grant DE-SC0011074, National Science Foundation
Grant AGS-1505198, National Oceanic and Atmospheric Administration Grant
NA14OAR4310274, and a Dissertation Year from the University of California, Los Angeles
Fellowship (KS). Parts of this material have been presented at the Fall 2016 meeting of the
American Geophysical Union and have formed part of K. Schiro's PhD thesis. We thank S.
Giangrande for providing pre-processed radar wind profiler data and for helpful discussions.



















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

**Figures**

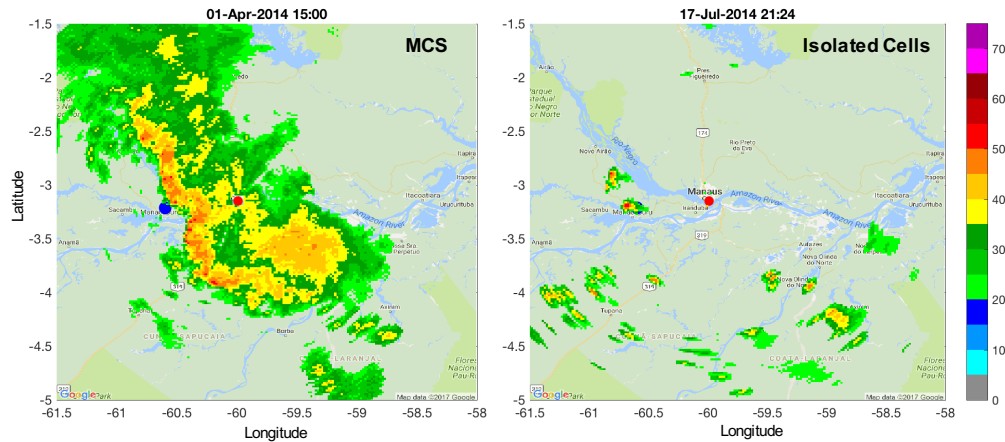


**Figure 1: Examples from S-Band Radar on 01 Apr 2014 at 15:00 UTC (11:00 LT) before**
**the passage of an MCS, and at 17 Jul 2017 at 21:24 UTC (17:24 LT) after the passage of an**
**isolated cell. The red dot indicates the location of the S-Band radar, and the blue dot**
**indicates the location of the main GOAmazon site (T3).**

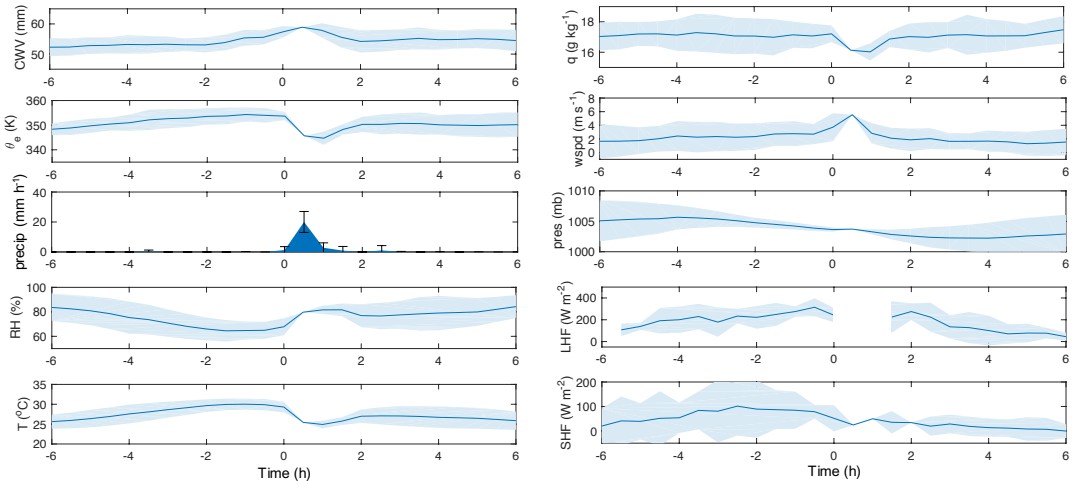


**Figure 2: Composites of meteorological variables from the AOSMET station at site T3 6 h**
**leading and 6 h lagging the 30-minute interval right before the drop in equivalent potential**
**temperature (2nd panel) and precipitation maximum (3rd panel) coincident with the passage**
**of isolated cells. Error bars are +/- 1 standard deviation with respect to 0.5 h.**



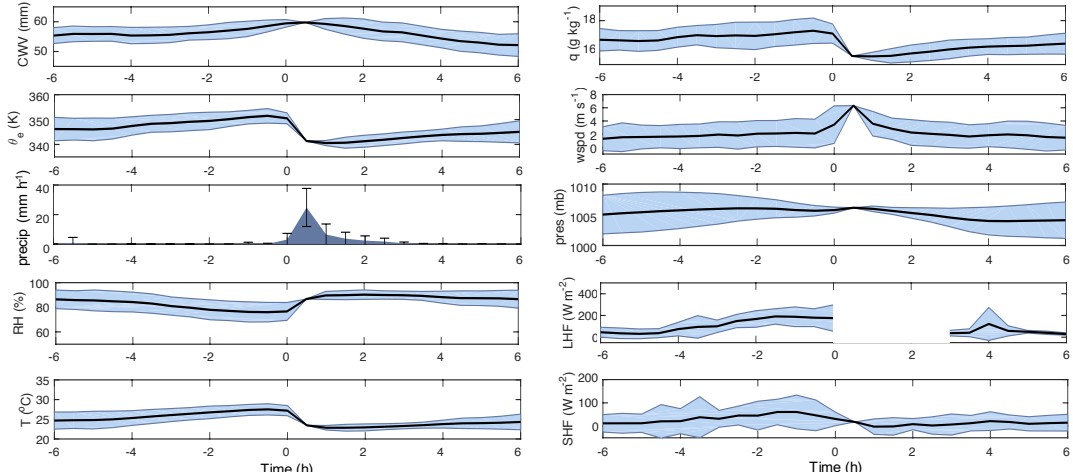


**Figure 3: Same as Fig. 2, except for MCSs.**

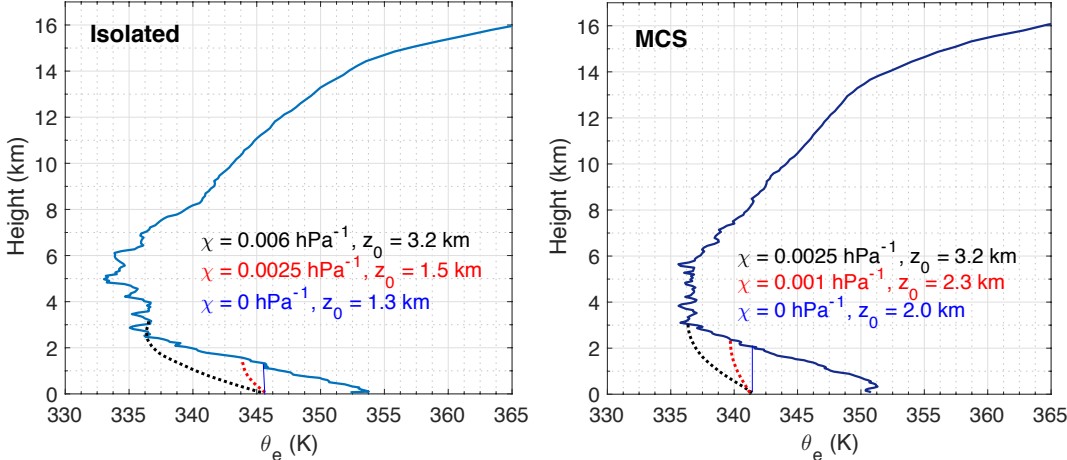


**Figure 4: Mean profiles of $\theta_e$ for isolated cells (left) and MCSs (right) within 6 h leading**
**the passage of a deep convective event. Dashed lines indicate the mean descent path for**
**plumes originating at various altitudes and mixing with the environment at various rates;**
**solid blue line shows mean descent without mixing.**




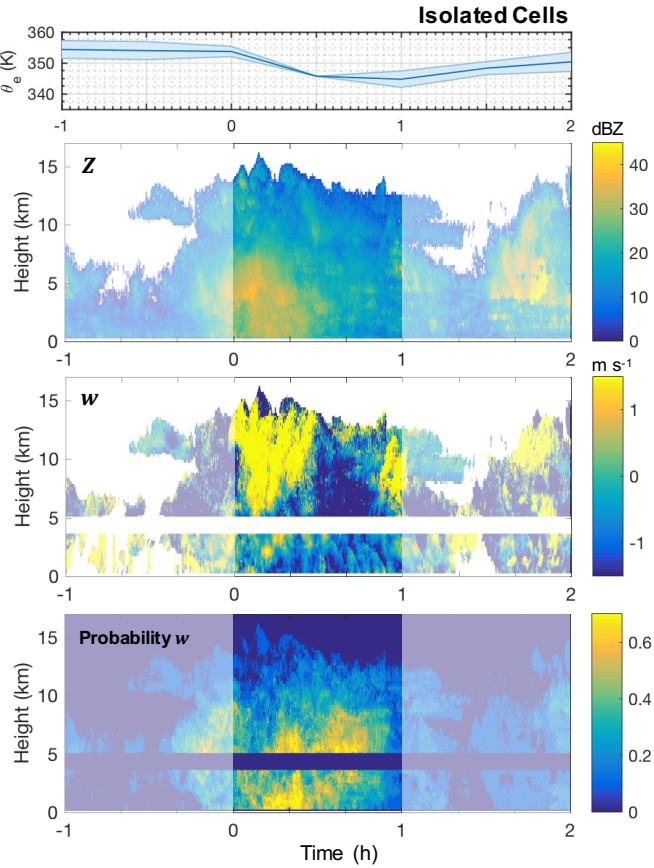


**Figure 5: The composite $\theta_e$ surrounding minimum $\Delta\theta_e$, as in Fig. 2 (top panel), mean**

**reflectivity (dBZ; second panel), mean vertical velocity (third panel; m s$^{-1}$), and probability**

**of w < 0 m s$^{-1}$ (bottom panel) measured by the radar wind profiler at T3 leading and**

**lagging the passage of isolated cells.**





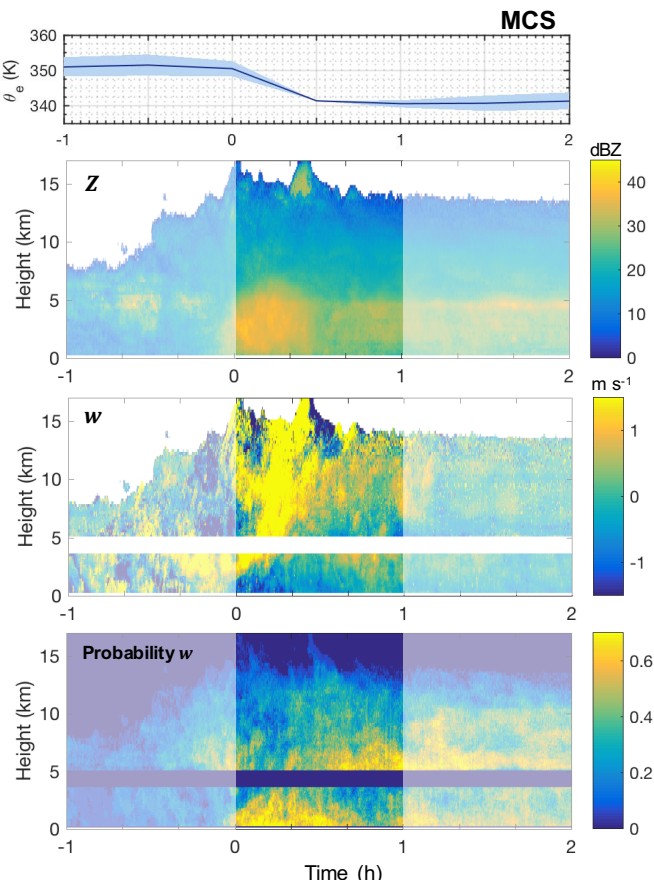


**Figure 6: Same as Fig. 5, but leading and lagging the passage of MCSs.**




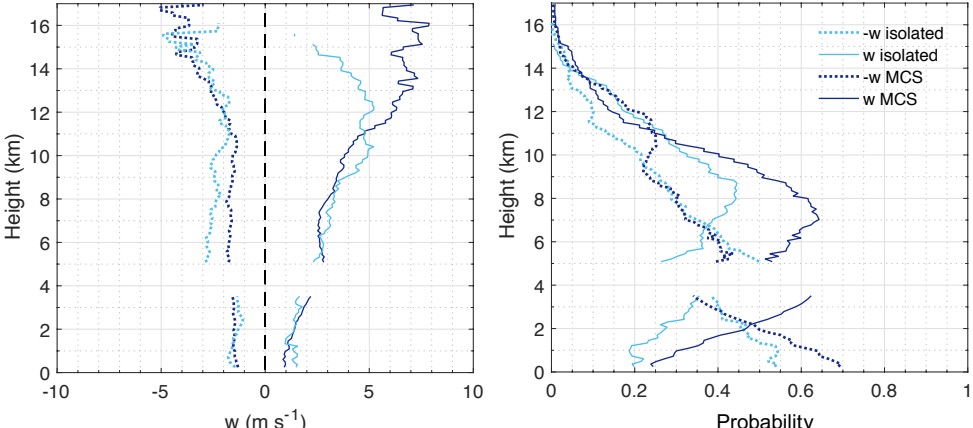


**Figure 7: (left) Mean vertical velocity profiles for MCSs and isolated cells for downdrafts (w < 0 m s⁻¹; dashed) and updrafts (w > 0 m s⁻¹; solid). (right) Mean probability of observing updrafts or downdrafts as a function of altitude. Means are composited from data in the 30 minutes of largest drop in $\Delta\theta_e$ (0-0.5 h in Figs. 2, 3, 5, and 6).**

**(left) Mean vertical velocity profiles for MCSs and isolated cells for downdrafts**
**(w < 0 m s⁻¹; dashed) and updrafts (w > 0 m s⁻¹; solid). (right) Mean probability of**
**observing updrafts or downdrafts as a function of altitude. Means are composited from**
**data in the 30 minutes of largest drop in $\Delta\theta_e$ (0-0.5 h in Figs. 2, 3, 5, and 6).**

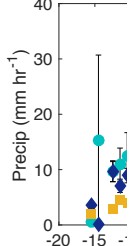
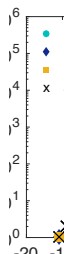


**Figure 8: (left) Precipitation (30-min averages) conditionally averaged by coincident changes in equivalent potential temperature ($\Delta\theta_e$) at the GOAmazon site. Precipitation values corresponds to the $\theta_e$ values at the end of each differencing interval. Bins are a width of 1°. (middle) The probability of precipitation (> 1 mm h⁻¹) occurring for a given $\Delta\theta_e$. (right) The frequency of occurrence of $\Delta\theta_e$ and precipitation for a given $\Delta\theta_e$ (precip > 1 mm h⁻¹). Precipitation derived from S-Band radar reflectivity at spatial averages over 25 km and 100 km grid boxes surrounding the GOAmazon site are included for comparison to the in situ precipitation.**

**(left) Precipitation (30-min averages) conditionally averaged by coincident**
**changes in equivalent potential temperature ($\Delta\theta_e$) at the GOAmazon site. Precipitation**
**values corresponds to the $\theta_e$ values at the end of each differencing interval. Bins are a**
**width of 1°. (middle) The probability of precipitation (> 1 mm h⁻¹) occurring for a given**
**$\Delta\theta_e$. (right) The frequency of occurrence of $\Delta\theta_e$ and precipitation for a given $\Delta\theta_e$ (precip >**
**1 mm h⁻¹). Precipitation derived from S-Band radar reflectivity at spatial averages over 25**
**km and 100 km grid boxes surrounding the GOAmazon site are included for comparison to**
**the in situ precipitation.**



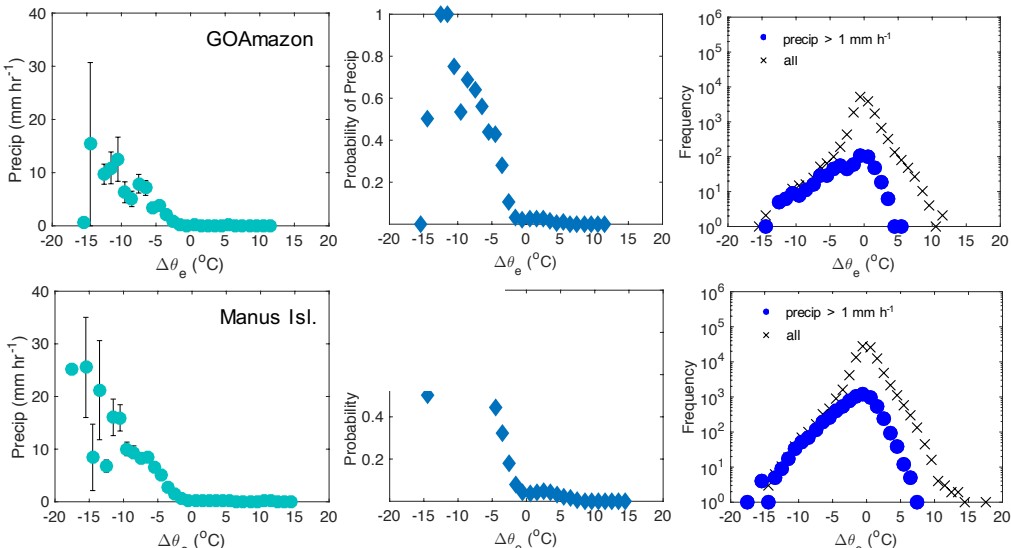


**Figure 9: (left) Precipitation (30-min averages) conditionally averaged by coincident changes in equivalent potential temperature ($\Delta\theta_e$) at the GOAmazon site (top) and Manus Island (bottom). Precipitation values corresponds to the $\theta_e$ values at the end of each differencing interval. Bins are a width of $1^o$. (middle) The probability of precipitation occurring for a given $\Delta\theta_e$. (right) The frequency of occurrence of $\Delta\theta_e$ and precipitation for a given $\Delta\theta_e$.**

705



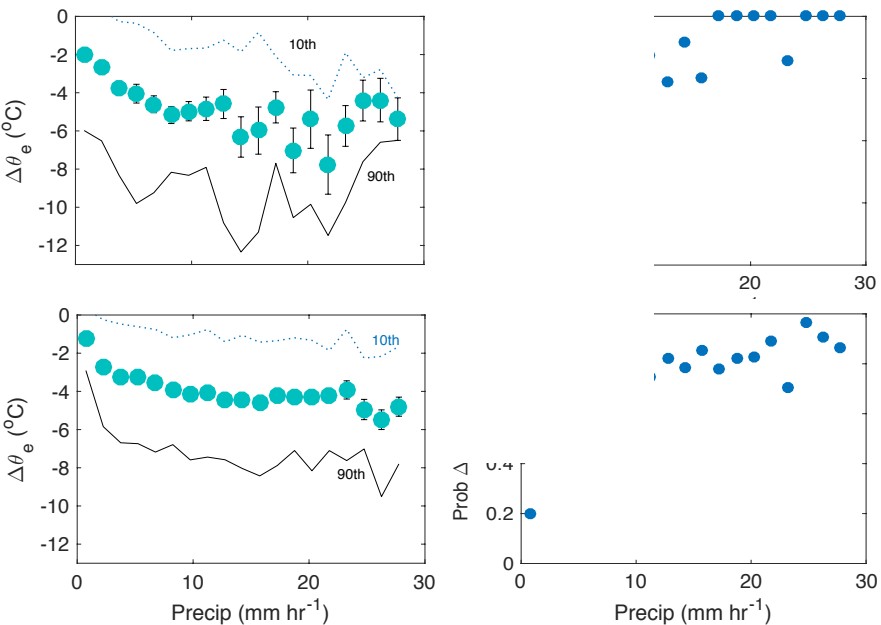

706

**Figure 10: $\Delta\theta_e$ conditionally averaged by coincident precipitation (1-h averages) at the**

**GOAmazon site (top) and at Manus Island (bottom). Precipitation values corresponds to**

**the $\theta_e$ values at the end of each differencing interval. Bins are a width of $1^o$. Error bars**

**represent standard error. The 10th and 90th percentile values for each bin are drawn for**

**reference.**

712