# Peer review of "Tropical Continental Downdraft Characteristics: Mesoscale Systems versus Unorganized"

_Atmospheric Chemistry and Physics, 2017_

## Referee Comment (RC1) · Anonymous Referee #2 · 15 Sep 2017

Review of

"Tropical Continental Downdraft Characteristics: Mesoscale Systems versus Unorganized Convection"

by Schiro and Neelin

Atmospheric Chemistry and Physics

15 September 2017

This paper uses the observations of the GOAmazon campaign to characterize the downdrafts and contrast the characteristics of the Mesoscale Systems downdrafts

versus the unorganized convection downdrafts. Time evolutions of thermodynamical characteristics around the moment of minimum equivalent potential temperature are shown and compare well with the literature. Then, composite of vertical velocity and reflectivity profiles are realised for both isolated cells and MCS. I am surprised by the un-smoothness of the figures (the same for the vertical profile in Fig 4) even though averaged over more than 10 cases. The highest probability of downdrafts in the levels below 3km is in agreement with the literature. The last section analyses the relationship between the drop in equivalent potential temperature and the precipitation. It could be improved by adding a discussion on how this relationship could be used as a constraint for parameterization. This paper is relatively well written even though some work on figures is still needed. However, I found that it does not present very original results. I suggest to better discussed or put into context the results. I have a few comments below that the authors should consider in revising their manuscript.

Comments:

Abstract: -l 18 : change 'have a probability' to 'have a probability of occurrence' -The abstract is too long. Please reduce it. I propose to suppress the sentence from l 20 to l 23. -You should also reformulate the end of the abstract after l28 which is unclear.

Introduction: This section could refer more to previous studies that use observations to document cold pools. Here are a few examples of reference missing: Charba, 1974; Engerer et al, 2008; Feng et al 2015; Redl et al 2015. Some are quoted in the rest of the text but this section should provide an overview of what we know about downdrafts from observations.

Data & Methods: -L 115: 'that create a subsequent drop in $\theta$ " at the surface of less than -5 o C': (also line 140 and line 167) How is the drop quantified ? Over which time interval? Please be more specific. From Fig 2, it seems that the drop is more than -5°C. And This is also indicated in l148 to 150: 'Values of theta_e are 353.6K on average before passage of the cell. An hour after the passage, the theta_e value

Interactive
comment

drops by an average of 8.9° to an average value of 344.7K' -Also please mention here than no latent heat flux measurements are available at the passage of the convective systems and the following hour. -Please indicate how the recovery time is computed. Surface thermodynamics: - l 174-175: please indicate also the average value of CWV for the isolated cells. For the MCS, please compare the figure showing the evolution of the CWV with Figure 3 of Taylor et al, 2017 The surface flux panels are not commented in the text so either removed them from Fig 2 and 3 or comment them.

Downdraft origin and the effects of mixing: - the 1.3km and 2km for the origin of the downdraft with the assumption of no-mixing is only derived for one given case? Or is obtained from the composite of all the radiosondes for a given category? Could you please comment on the range of values obtained for all the cases and change Figure 4 by one figure showing the mean profile +/- the standard deviation.

Vertical velocity and downdraft probability: - This section is a bit short on conclusion ('These results suggest that in most downdrafts, a substantial fraction of the air reaching the surface originates in the lowest 3km'). According to Figure 7, above 3km there is still a probability of ∼0.4 to get a downdraft. Also, Fig 5, 6 and 7 do not show much details in the lower levels you may want to zoom this figure in the lower 7 or 10 km. - please comment on the dispersion on reflectivity and vertical velocity obtained for the different individual cases. Relating cold pool thermodynamics to precipitation - l 328 change 'in Fig 5' to 'in Fig 4' - How do you interpret the fact that you have some points with very large negative anomaly (<15°) in equivalent potential temperature and no precipitation? - Please indicate that the Dtheta_e is always the same one, i.e. determined from in-situ observations. - l357-359: minimum or maximum of Dtheta_e: please make it clearer. :'The MAXIMUM of Dtheta_e within a 3-h window of a given precipitation rate is averaged to minimize the effects of local precipitation maxima occurring slightly before or after the MINIMUM in Dtheta_e'. -Please detail more how those diagnostics could be used as a constraint for parameterization.

Conclusions: - Temperature drop of 3.9°C or 4.4° and Equivalent potential temperature from of 8° or 8.9° (line 154 or line 381) for isolated cells: please be consistent. Also check the values for the MCS cases: they are in consistent in the text and the conclusion - 'with the moisture recovering faster than temperature': do you have an assumption to explain such feature. - L 411: 'For area-averaged precipitation on scales typical of GCM grids, precipitation magnitude is lower for strong, negative Dtheta_e, consistent with the points with large D_theta_e occurring at localized downdraft locations within a larger system with smaller area-average precipitation': I don't get the argument: why the Dtheta_e will not also be smaller in this case?

Figures: - Figure 1: Please provide in the caption the name of the field that is drawn and its unity - Figure 2: It will help the lecture of this graph if there were some horizontal lines for the values as shown in the upper panel of Figure 5. Otherwise, it is very difficult to get a quantitative information from these subplots. In the caption, you mention overbars, in fact there are only drawn for the precipitation; for the other parameters, a shading is indicated around the mean: please modify accordingly the caption. - Figure 3: to help in the comparison please add the mean values of Figure 2 on Figure 3 with a dashed line. - Figure 4: I guess those profiles are from one radiosonde profile only and a given case for each case (otherwise I do not expect such small scale vertical variations of the equivalent potential temperature for an average over more than 10 radiosonde profiles). Please replace by a figure showing the mean and +/- the standard deviation shown by a shading. - Figures 5 and 6 : I am impressed by the relatively un-smooth aspect of those figures for an average over 11 and 17 cases respectively. For the vertical velocity please use a red-blue colour bar in order for the reader to more rapidly identify the ascending versus descending areas. -Figure 8: what is the unit of the right panel. Please keep the same colour legend for all subplots. - Figure 9 & 10 : please add both sites on the same sub-plots and reduce the number of subplots from 3 to 6.

References: Charba, J., 1974: Application of gravity current model to analysis of squall-line gust front. Mon. Weather Rev., 102, 140–156 Redl, R., A. H. Fink, and P. Knippertz, 2015: An Objective Detection Method for Convective Cold Pool Events and Its Application to Northern Africa. Mon. Weather Rev., 143, 5055–5072 Taylor C.M., D Belusic, F Guichard, D J Parker, T Vischel, O Bock, P P Harris, S Janicot, C Klein, G Panthou, 2017Âă: Frequency of extreme Sahelian storms tripled since 1982 in satellite observations. Nature, 544, 475-478

---

## Referee Comment (RC2) · Anonymous Referee #3 · 19 Sep 2017

This paper uses GOAmazon data to compare characteristics of downdrafts in unorganized convection and those of Mesoscale Convective Systems (MCSs). Using composites of meteorological variables, the paper provides a quantification of the changes in surface thermodynamic fields associated to the passage of the observed convective events and their relative cold pools. The paper then focuses on the properties of downdrafts using surface values of equivalent potential temperature and composites of radar reflectivity and vertical velocity. It is shown that downdrafts have similar intensity and originate in the lower troposphere, with MCSs' downdrafts having a slightly higher origin height. Finally, a strong relationship is shown between drops in equivalent potential temperature at the surface and precipitation rate.

The paper is overall well written and presents interesting results from observations. I only have few comments that I list below, but otherwise I recommend its publication in this journal.

Comments:

Line 40: I believe the surname is actually Böing. Line 41-42: The boundary between the cold pool and the environment is not, strictly speaking, a mechanism. Please rephrase this. Line 149 and following: I find the use of Celsius and Kelvin at the same time confusing. Please use Kelvin throughout the manuscript. Line 149: Please specify units of measurements for 8.9. Line 164: Could the greater recovery of the temperature be simply due to the diurnal cycle (i.e., the fact that some of the systems you are observing are in the late part of the day)? Line 187: I think "corresponding" would be a better term here. Line 219-220: Judging from Figure 4, the minimum of theta_e for the isolated case seems much higher than what you indicated, more like 5 km. Line 224-229: Could you speculate whether a higher mixing rate for isolated convection would actually make sense? Line 248-250: You say that retrieval near freezing level has large errors, so how confident are you about the high probability you mention? Line 286: Betts 1976 should have parentheses? Line 292: The altitude of 1.5 km is cited only as a reference point. The mode of the distribution seems to actually be at 1 km. Line 318: The relationship in Figure 8 seems non-linear with a plateau/decrease towards lower dtheta_e. Why is that? Line 356: "The maximum dtheta_e [. . .]". Do you mean the minimum? Line 379: Please check the number you are providing here as they don't seem in agreement with what you reported earlier on. Line 381-383: Why is moisture recovering faster? Line 395-396: Again, it would be very nice if you could suggest reasons why this could happen. Line 419: Do you mean "composing"?

Figure 1: Please specify unites of measurement on top of the color bar. Figure 2-3: As they are, these figures make it hard to appreciate details of the curves. First, I would recommend including a grid in each single panel; second, I would also suggest using a lower aspect ratio so that panels are less squeezed.

[Figure]

---

## Referee Comment (RC3) · Anonymous Referee #1 · 25 Sep 2017

Review of:

Tropical continental downdraft characteristics: mesoscale systems versus unoganized convection, by K. Schiro and J. D. Neelin

This manuscript assesses the cold pool characteristics associated with organized and unorganized convection over the Amazon, as inferred from GoAmazon campaign data. A valuable inclusion is radar wind profiler measurements of vertical velocity, and assessments of common mixing paradigms. This is an interesting article and I only have minor comments. My most major comment is that I had trouble seeing the same features in Figs 5 and 6 that the authors mention in the main text. I think it is just a matter

of re-drafting the figures.

1. abstract: it's worth mentioning that the analysis was focused on the more extreme convection, consisting of 11 isolated cells and 17 MSCs. A sentence discussing the differences between the isolated and organized convection cases would also be useful.

2. line 88: 30-minutes strikes me as a long time span over which to average cold pool changes, which those changes easily happening over shorter time spans. Why did the authors choose this time scale? can they say something here about the ability to resolve temporal evolution? on line 137 you mention averaging over 1 hour, even longer.

3. were all of the cold pools preceeded by unmodified conditions? cold pools tend to cluster.

4. discussion of Figs 2 and 3: do the individual examples all follow the same evolution as is shown for the mean composite?

5. lines 199-200, fig. 3: it is difficult to discern a difference of 700m between 2 separate plots extending up to 17km. I would encourage the authors to try out different plotting formats, perhaps one plot showing both of the mean profiles together up to 17km, and another one zoomed in to the 0-4km range would work, showing all 6 mixing lines. This would help with interpretation of the mixing rates and their differences for the two forms of convection, as discussed in lines 215-225, as well.

6. line 242-243: it is difficult to see the downdraft this sentence is referring to in Fig. 5. perhaps an arrow, or a color scheme emphasizing the stronger downdrafts, would help. the latter might be my suggestion, to use e.g. red for downdrafts less than -1 m/s and yellow for updrafts > 1 m/s. or vice versa, in which case you might have something that relates well to the probability of downdrafts figure in the bottom panel.

I also wonder if it would be useful to blow up the 0-4km altitude range in Figs 5 and 6. The manuscript makes the argument that downdrafts originate from the lower free

troposphere, but these figures focus the eye on the upper troposphere. I have trouble distinguishing features mentioned in the text (e.g., lines 260-261) in the figures. One idea might be to make this 6-paneled figures with 3 additional panels added per figure that focus on the 0-5km range.

7. p. 9: I see no discussion of wind shear here. What role if any does the (horizontal) wind profile play in this? line 260-261 would suggest none, is this consistent with conceptual views of MCS organization?

8. line 256: how can downdraft air be positively buoyant? does it overshoot its level of neutral buoyancy?

9. line 263: I have trouble distinguishing this feature. is this occurring between 1-2 hours near the surface?

10. lines 283-296: see also de Szoeke et al 2017 JAS for further corroborating observations from DYNAMO.

lines 359-361: I wonder if sampling can explain why you might find a strong precipitation event without a decrease in surface theta-e, as it doesn't quite make sense to me that this would be the case, unless the decrease in surface theta-e is simply displaced.

minor comments:

line 45: Zuidema et al 2011 should be Zuidema et al 2012 line 71: provides should be provide line 140-141: the language here is slightly unclear ("drops of -5C or less"). would suggest referencing to an absolute value. line 167: typo at end line 257: mention the gravity waves are in the stratosphere

reference: de Szoeke et al, 2017: Cold pools and their influence on the tropical marine boundary layer. J. Atmos. Sci.,74, pp. 1149-1167, doi:10.1175/JAS-D-16-0264.1

Please also note the supplement to this comment:
https://www.atmos-chem-phys-discuss.net/acp-2017-684/acp-2017-684-RC3-

supplement.pdf

---

## Author Comment (AC1) · 6 Dec 2017

Responses to reviewers for "Tropical Continental Downdraft Characteristics: Mesoscale Systems versus Unorganized Convection" by K. A. Schiro and J. D. Neelin

Reviewer #1

1. It's worth mentioning that the analysis was focused on the more extreme convection, consisting of 11 isolated cells and 17 MCSs. A sentence discussing the differences between the isolated and organized convection cases would also be useful.

Thanks, in the abstract line 11 we added the word "strong" to clarify. Additionally, lines 14-17 were modified to elaborate on similarities and differences between MCSs and isolated cells.

2. line 88: 30-minutes strikes me as a long time-span over which to average cold pool changes, which those changes easily happening over shorter time spans. Why did the authors choose this time scale? can they say something here about the ability to resolve temporal evolution? on line 137 you mention averaging over 1 hour, even longer.

We originally chose 30 minutes for these types of composites because otherwise it is difficult to condition on a distinct decrease in $\theta_e$. If conditioning on, say, 5 min average decreases in $\theta_e$, the decrease in $\theta_e$ observed over that period of time may not be unlike usual fluctuations in $\theta_e$ throughout the diurnal cycle. On the other hand, large decrease in $\theta_e$ over a 15 or 30 minute timeframe are frequently attributable to the passage of a cold pool. We have modified Fig. 2 to include composites of 5 min average quantities, yet we still condition on the decreases in $\theta_e$ of >=5K and precipitation rates >= 10 mm/hr over a 30 minute time period, as was originally the case for the aforementioned reason.

3. were all of the cold pools preceded by unmodified conditions? cold pools tend to cluster.

It is difficult to say with certainty whether the cold pools were preceded by unmodified conditions; the composites do not show any significant precipitation occurring beforehand, however, and thus we do not suspect that the surface thermodynamics had been appreciably modified by precipitation or cold pools.

4. discussion of Figs 2 and 3: do the individual examples all follow the same evolution as is shown for the mean composite?

The evolution can vary to some degree. We try to illustrate this evolution with shading (+/- 1 std. dev.), which are calculated with respect to the value of minimum $\theta_e$ within a cold pool (discussed in lines 131-133). Additionally, for visual clarity, we shifted the minimum $\theta_e$ to time 0 in Fig. 2.

5. lines 199-200, fig. 3: it is difficult to discern a difference of 700m between 2 separate plots extending up to 17km. I would encourage the authors to try out different plotting formats, perhaps one plot showing both of the mean profiles together up to 17km, and another one zoomed in to the 0-4km range would work, showing all 6 mixing lines. This would help with interpretation of the mixing rates and their differences for the two forms of convection, as discussed in lines 215-225, as well.

Thanks for this suggestion. We restricted the height to 7 km to zoom in on the features in the lowest levels more closely.

6. line 242-243: it is difficult to see the downdraft this sentence is referring to in Fig. 5. perhaps an arrow, or a color scheme emphasizing the stronger downdrafts, would help. the latter might be my suggestion, to use e.g. red for downdrafts less than -1 m/s and yellow for updrafts > 1 m/s. or vice versa, in which case you might have something that relates well to the probability of downdrafts figure in the bottom panel.

I also wonder if it would be useful to blow up the 0-4km altitude range in Figs 5 and 6. The manuscript makes the argument that downdrafts originate from the lower free troposphere, but these figures focus the eye on the upper troposphere. I have trouble distinguishing features mentioned in the text (e.g., lines 260-261) in the figures. One idea might be to make this 6-paneled figures with 3 additional panels added per figure that focus on the 0-5km range.

We added two additional panels in Figs. 4-5 (old Figs. 5-6) to zoom in on the 0-4 km region to emphasize the downdrafts being discussed in the text.

7. p. 9: I see no discussion of wind shear here. What role if any does the (horizontal) wind profile play in this? line 260-261 would suggest none, is this consistent with conceptual views of MCS organization?

We did not evaluate the effects of wind shear on MCS organization and downdrafts, although it is known to play an important role. Our goal here was to estimate downdraft origin height for the downdrafts associated with the initial sharp drop in $\theta_e$ at the surface. We would expect that wind shear is important to these MCSs and their downdrafts, as moderate shear is known to be favorable for the development of MCSs (e.g. Wesiman and Klemp 1982; Rotunno et al. 1988), but we cannot comment on the effects of wind shear on updraft or downdraft properties from the analysis presented.

8. line 256: how can downdraft air be positively buoyant? does it overshoot its level of neutral buoyancy?

This discussion is referring to the results in Sun et al. (1993). Because they are a dynamical response to the updraft, they are dynamically pushed downwards, but would otherwise be thermodynamically unstable (i.e., positively buoyant). See Sun et al. (1993) for a comprehensive discussion.

9. line 263: I have trouble distinguishing this feature. is this occurring between 1-2 hours near the surface?

We have modified this sentence accordingly in lines 276-279, as it is a bit difficult to see from the composites (although they more than likely exist).

10. lines 283-296: see also de Szoeke et al 2017 JAS for further corroborating observations from DYNAMO.

Thank you. We added this reference to support the discussion in line 309.

11. lines 359-361: I wonder if sampling can explain why you might find a strong precipitation event without a decrease in surface $\theta_e$, as it doesn't quite make sense to me that this would be the case, unless the decrease in surface $\theta_e$ is simply displaced.

Displacement is possible, but the results presented suggest that it does not greatly affect the results. For instance, we tested to see whether $\theta_e$ is displaced from precipitation (e.g. large decreases in $\theta_e$ without appreciable precipitation) using radar data in Fig. 7. This does not modify the distribution of precipitating points as a function of $\Delta\theta_e$ or the probability curve very much, suggesting that the in situ $\Delta\theta_e$ and precipitation correspond well to one another (we commented on this in lines 364-366). So it is likely, as shown in the distribution of precipitating points and other evidence from Figs. 3-6 that downdrafts commonly originate at low enough levels where the decrease in $\theta_e$ is small (or they mix considerably enough to make the decrease negligible).

Another thing to consider is displacement in time. In Figs. 7 and 8, the statistics are compiled based on 1-h average values of $\Delta\theta_e$ and precipitation. When we bin and conditionally average the precipitation by $\Delta\theta_e$ we only condition based on events occurring within that hour. The 1-h interval was chosen to hopefully be a coarse enough resolution to capture the $\Delta\theta_e$ from a cold pool in one time step, yet a high enough resolution to retain the signal despite averaging. We suspected, however, that this interval might not be wide enough to capture all precipitation that falls with each system in any given hour. So in Fig. 9, we try to avoid this as best as possible by instead conditioning on the maximum precipitation rates and minimum $\Delta\theta_e$ (1-hr average values) within a given 3 hour interval. Nevertheless, from what we can tell, the results do not appear very sensitive to displacements in space or time, thus confirming that there are modest $\Delta\theta_e$ decreases that occur without coincident precipitation (and vice versa) and that displacement is not playing a major role.

There is also some dependence on the threshold $\theta_e$ decrease chosen to define cold pools (e.g. in Fig. 9 and in the stats presented in lines 409-414), though we feel that -2 K is appropriate. The variance of $\Delta\theta_e$ is 3.12 (mean is ~0) and the standard deviation is 1.76, so we chose -2 to be below 1 $\sigma$.

12. line 45: Zuidema et al 2011 should be Zuidema et al 2012

Thanks. This has been corrected.

13. line 71: provides should be provide

Corrected. Please see tracked changes.

14. line 140-141: the language here is slightly unclear ("drops of -5C or less"). would suggest referencing to an absolute value.

Corrected. Please see tracked changes.

15. line 167: typo at end

Corrected. Please see tracked changes.

16. line 257: mention the gravity waves are in the stratosphere

Corrected. Please see tracked changes.

Reviewer #2:

1. (Abstract) line 18 : change 'have a probability' to 'have a probability of occurrence' -The abstract is too long. Please reduce it. I propose to suppress the sentence from l 20 to l 23. -You should also reformulate the end of the abstract after l28 which is unclear.

We have reduced the abstract and clarified as needed, per this suggestion. Please see tracked changes for specifics.

2. (Introduction) This section could refer more to previous studies that use observations to document cold pools. Here are a few examples of reference missing: Charba, 1974; Engerer et al, 2008; Feng et al 2015; Redl et al 2015. Some are quoted in the rest of the text but this section should provide an overview of what we know about downdrafts from observations.

We have added some additional references to this section, as suggested.

3. (Data & Methods) Line 115: 'that create a subsequent drop in $\theta_e$ at the surface of less than -5 C': (also line 140 and line 167) How is the drop quantified? Over which time interval? Please be more specific.

Thanks for pointing this out. The drops are quantified within a 30-min interval. The $\theta_e$ from the previous interval is subtracted from the following interval, which defines the $\Delta\theta_e$. If that $\Delta\theta_e$ is less than –5 K (and precipitation is greater than 10 mm hr$^{-1}$) we include the convective event in our composite. We clarified this in lines 103-105.

4. From Fig 2, it seems that the drop is more than -5$^{\circ}$C. And This is also indicated in l148 to 150: 'Values of $\theta_e$ are 353.6K on average before passage of the cell. An hour after the passage, the $\theta_e$ value drops by an average of 8.9$^{\circ}$ to an average value of 344.7K'

Yes, we condition on events with depressions greater than 5 K, which is why the composites show values greater than 5 K.

5. Also please mention here than no latent heat flux measurements are available at the passage of the convective systems and the following hour.
6. The surface flux panels are not commented in the text so either removed them from Fig 2 and 3 or comment them.

We have removed the surface flux panels in the main text, with brief discussion about this issue, and have moved them to the SI (Figs. S1 and S2).

7. Please indicate how the recovery time is computed.

Yes, thanks. Please see tracked changes (lines 135-138).

8. (Surface thermodynamics) lines 174-175: please indicate also the average value of CWV for the isolated cells. For the MCS, please compare the figure showing the evolution of the CWV with Figure 3 of Taylor et al (2017).

We added the values for more quantitative comparison, as suggested, and have compared to Taylor et al. (2017).

9. (Downdraft origin and the effects of mixing) The 1.3km and 2km for the origin of the downdraft with the assumption of no-mixing is only derived for one given case? Or is obtained from the composite of all the radiosondes for a given category? Could you please comment on the range of values obtained for all the cases and change Figure 4 by one figure showing the mean profile +/- the standard deviation.

This is the mean of all profiles in a given category, and we compute the mixing from the mean profile. We added to the text in lines 202-206 for clarification. We also modified the figure to add error bars (+/- 1 standard error).

10. (Vertical velocity and downdraft probability) This section is a bit short on conclusion ('These results suggest that in most downdrafts, a substantial fraction of the air reaching the surface originates in the lowest 3km'). According to Figure 7, above 3km there is still a probability of ~0.4 to get a downdraft.

We have modified this wording in lines 305-307 as follows: "These results, and those presented in the previous section, suggest a range of downdraft origin levels throughout the lowest few kilometers within both organized and unorganized convective systems."

11. Also, Fig 5, 6 and 7 do not show much details in the lower levels you may want to zoom this figure in the lower 7 or 10 km.

We have zoomed Figs. 3-5 per this suggestion.

12. please comment on the dispersion on reflectivity and vertical velocity obtained for the different individual cases.

We rearranged and added to the discussion surrounding Fig. 6 to help clarify the methods, and added a discussion in lines 300-304 to place bounds on the variability observed.

13. line 328 change 'in Fig 5' to 'in Fig 4' –

Thanks; this has been corrected.

14. How do you interpret the fact that you have some points with very large negative anomaly (<15°) in equivalent potential temperature and no precipitation?

We have since eliminated these points from the figures since we tightened the constraints on the counts required to produce a robust signal (minimum counts: 5).

15. Please indicate that the $\Delta\theta_e$ is always the same one, i.e. determined from in-situ observations.

A sentence was added (line 352-353) to clarify this.

16. l357-359: minimum or maximum of $\Delta\theta_e$: please make it clearer. :'The MAXIMUM of Dtheta_e within a 3-h window of a given precipitation rate is averaged to minimize the effects of local precipitation maxima occurring slightly before or after the MINIMUM in $\Delta\theta_e$'.

Thanks for pointing this out. We have revised this wording. Please see tracked changes.

17. Please detail more how those diagnostics could be used as a constraint for parameterization.

Thanks for the suggestion. We elaborated on this in lines 405-414.

18. (Conclusions) Temperature drop of 3.9$^\circ$C or 4.4$^\circ$ and Equivalent potential temperature from of 8$^\circ$ or 8.9$^\circ$ (line 154 or line 381) for isolated cells: please be consistent. Also check the values for the MCS cases: they are in consistent in the text and the conclusion –

We have checked this for consistency and reported the correct values in both places. Thanks for pointing this out.

19. 'with the moisture recovering faster than temperature': do you have an assumption to explain such feature.

We suspect that after the storm passes, the persistence of clouds hinders an immediate increase of heating, yet increased evaporation from the wet surface can increase the moisture content. We added a brief discussion to lines 157-161 to address this.

20. L 411: 'For area-averaged precipitation on scales typical of GCM grids, precipitation magnitude is lower for strong, negative $\Delta\theta_e$, consistent with the points with large $\Delta\theta_e$ occurring at localized downdraft loca- tions within a larger system with smaller area-average precipitation': I don't get the argument: why the $\Delta\theta_e$ will not also be smaller in this case?

The $\Delta\theta_e$ we use is in situ; we do not have any spatial information. The area-averaged precipitation decreases with increasing area, but we don't have the ability to scale up the $\Delta\theta_e$.

21. (Figure 1) Please provide in the caption the name of the field that is drawn and its unity

Corrected. The field is reflectivity (dBZ).

22. (Figure 2) It will help the lecture of this graph if there were some horizontal lines for the values as shown in the upper panel of Figure 5. Otherwise, it is very difficult to get a quantitative information from these subplots. In the caption, you mention overbars, in fact there are only drawn for the precipitation; for the other parameters, a shading is indicated around the mean: please modify accordingly the caption.

We have modified the figure according to your suggestion.

23. (Figure 3) to help in the comparison please add the mean values of Figure 2 on Figure 3 with a dashed line.

Thanks for this suggestion. We combined Figs. 2 and 3 based on this recommendation.

24. (Figure 4) I guess those profiles are from one radiosonde profile only and a given case for each case (otherwise I do not expect such small scale vertical variations of the equivalent potential temperature for an average over more than 10 radiosonde profiles). Please replace by a figure showing the mean and +/- the standard deviation shown by a shading.

These are means of the MCSs and isolated events. We have added error bars (+/- 1 standard error).

25. (Figures 5 and 6) I am impressed by the relatively un-smooth aspect of those figures for an average over 11 and 17 cases respectively. For the vertical velocity please use a red-blue colour bar in order for the reader to more rapidly identify the ascending versus descending areas.

The plots do not appear smooth because we are showing composites of high-frequency radar wind profiler data, which can be somewhat noisy. The figure has been modified to help with this. After trying multiple color bars, it seemed clearest to keep the same color bar as before, but to add panels zoomed in on the lowest 4 km for visual clarity.

26. (Figure 8) What is the unit of the right panel? Please keep the same colour legend for all subplots. –

It is a frequency of the counts of $\Delta\theta_e$, and all colors should now be consistent (now Fig. 7).

27. (Figures 9 & 10) Please add both sites on the same sub-plots and reduce the number of subplots from 3 to 6.

Thanks for the suggestion. We have modified accordingly in Fig. 8 (old Fig. 9), which is very helpful. We also tried this for Fig. 9 (old Fig. 10), but the figure became too busy.

Reviewer #3

1. Line 40: I believe the surname is actually Böing.

Yes, thank you, we have modified this.

2. Line 41-42: The boundary between the cold pool and the environment is not, strictly speaking, a mechanism. Please rephrase this.

Thanks for pointing this out. Please see tracked changes.

3. Line 149 and following: I find the use of Celsius and Kelvin at the same time confusing. Please use Kelvin throughout the manuscript.

We modified all units to be Kelvin, as suggested, and have modified the figures and text accordingly.

4. Line 149: Please specify units of measurements for 8.9.

Corrected.

5. Line 164: Could the greater recovery of the temperature be simply due to the diurnal cycle (i.e., the fact that some of the systems you are observing are in the late part of the day)?

We think it is moreso the modification to the incoming solar insolation with cloud cover and added downdrafts from the stratiform region of MCSs that sets apart the recovery times for thermodynamic variables between the MCS and isolated cases. We believe this is the case because many of the MCSs are observed in the afternoon also, and inspection of the individual events does not lead us to believe it is an artifact of the diurnal cycle. We added a short discussion of this in lines 254-258.

6. Line 187: I think "corresponding" would be a better term here.

Thank you; we agree and modified the text accordingly.

7. Line 219-220: Judging from Figure 4, the minimum of $\theta_e$ for the isolated case seems much higher than what you indicated, more like 5 km.

We are careful not to claim where the exact level of the minimum in $\theta_e$ is, as it is a little difficult to tell in this composite and can range anywhere from ~3-7 km in height. We just simply choose a point at the same height as the MCS case at the lower end of this range (3.2 km). Additionally, since the values between 3-7 km (the mean values) are relatively similar, the mixing rate will not be sensitive to the exact level of origin.

8. Line 224-229: Could you speculate whether a higher mixing rate for isolated convection would actually make sense?

This is a very good question – one that also requires thought about the actual mixing paradigm. It depends on many factors, many of which are beyond the scope of this study. It is plausible that the environment mixes differently with the surrounding environment in isolated vs. MCS events, given differences in the dynamics between storm types. It is also possible that we are not optimally characterizing the vertical profile of the thermodynamic environment, since we are compositing radiosonde measurements within hours of the observed convective event, in addition to other associated errors (e.g., sample size). The main idea behind the analysis in Fig. 4 was to provide loose guidance for what seemed to be an appropriate degree of mixing given the current mixing paradigms employed operationally in GCMs. In response, we added to the discussion in lines 237-243 to help clarify our intention.

9. Line 248-250: You say that retrieval near freezing level has large errors, so how confident are you about the high probability you mention?

From a data quality perspective, we have greater confidence in convective regions than stratiform regions due to complex assumptions about microphysics and weaker air motions in stratiform regions. We thus restrict our main discussion and analysis to the convective regions. We have no reason to believe that we should not be confident in the quality of the retrievals in convective regions (S. Giangrande, personal communication).

10. Line 286: Betts 1976 should have parentheses?

Corrected.

11. Line 292: The altitude of 1.5 km is cited only as a reference point. The mode of the distribution seems to actually be at 1km.

Thanks. We've added "with the mode of the distribution nearer to 1 km" to line 332.

12. Line 318: The relationship in Figure 8 seems non-linear with a plateau/decrease towards lower $\Delta\theta_e$. Why is that?

This is a good question. It seems as though the higher rain rates do not necessarily correspond to colder cold pools. We are careful not to discuss this at length, as these data out to high $\theta_e$ are rare. We have since removed some of these data from the plots in Figs. 7-8, as they did not meet reasonable minimum count requirements.

13. Line 356: "The maximum $\Delta\theta_e$ [. . .]". Do you mean the minimum?

Yes, thanks. This has been corrected.

14. Line 379: Please check the number you are providing here as they don't seem in agreement with what you reported earlier on.

Thanks. We have confirmed/modified the values reported.

15. Line 381-383: Why is moisture recovering faster?
16. Line 395-396: Again, it would be very nice if you could suggest reasons why this could happen.

We suspect that after the storm passes, the persistence of clouds hinders an immediate increase of heating, yet increased evaporation from the wet surface can increase the moisture content. We added a brief discussion to lines 254-258 to address this.

17. Line 419: Do you mean "composing"?

It was supposed to be "compositing." Thanks for pointing this out; it has been corrected.

---

## Author Response (AR2)

Author's response #2:

One reviewer had the following additional comments:

- l 118 "All MCSs and isolated cells composited produce downdrafts that create a subsequent drop in the at the surface of more than 5 K in a 30-min period and have precipitation rates exceeding 10 mm h -1 within that same period." Isn't it clearer to directly state:
"every downdrafts, associated to either MCS or isolated cells, that create a subsequent drop in the at the surface of more than 5 K in a 30-min period and have precipitation rates exceeding 10 mm h -1 within that same period are composited"

Thanks. We have revised the wording to read, "Every downdraft associated with either MCSs or isolated cells that created a subsequent drop in $\theta_e$ at the surface of more than 5 K in a 30-min period and have precipitation rates exceeding 10 mm h$^{-1}$ within that same period are composited."

- l 273: mixing would need TO be 2 times greater.

Corrected.

In addition, I found a few more minor things:

- Line 134: 11 plus 6 would make 17 events. Please clarify here and throughout the manuscript the correct amount of events.

Thanks. We double-checked the numbers and corrected this error.

- Line 107: seconds should be abbreviated as 's'

Corrected.

- Line 98: Please double-check the correct full form of AOSMET (Aerosol Observing System Surface Meteorology?).

Thanks. This was corrected to Aerosol Observing System Surface Meteorology.

- Line 103: Please define LT at its first occurrence.

Corrected.

- Line 212: To be consistent use Kelvin here as well.

Corrected.

- Please double-check the correct numbering of the figures within the text.

Yes, we double-checked that the numbering is correct.

- Point 5 from Reviewer #2 was not specifically answered.

Our apologies. Point 5 was meant to be addressed below Point 6, as they were both related. We removed the discussion and analysis surrounding heat fluxes from the main text, and thus we no longer feel it is appropriate to add in this information in the suggested spot. We have, however, added the appropriate information in the supplement where we discuss heat fluxes.

[revised manuscript text omitted]